# Iron Status of Vegans, Vegetarians and Pescatarians in Norway

**DOI:** 10.3390/biom11030454

**Published:** 2021-03-18

**Authors:** Sigrun Henjum, Synne Groufh-Jacobsen, Tonje Holte Stea, Live Edvardsen Tonheim, Kari Almendingen

**Affiliations:** 1Department of Nursing and Health Promotion, Faculty of Health Science, Oslo Metropolitan University, Kunnskapsveien 55, 2007 Kjeller, Norway; live.tonheim@gmail.com (L.E.T.); kalmendi@oslomet.no (K.A.); 2Department of Nutrition and Public Health, Faculty of Health and Sport Sciences, University of Agder, Universitetsveien 25, 4630 Kristiansand, Norway; synne.groufh.jacobsen@uia.no; 3Department of Health and Nursing Sciences, Faculty of Health and Sport Sciences, University of Agder, Universitetsveien 25, 4630 Kristiansand, Norway; tonje.h.stea@uia.no

**Keywords:** ferritin, iron, iron status, micronutrients, pescatarians, plant-based diet, transferrin saturation, vegans, vegetarians

## Abstract

Although plant-based diets provide well-established physical and environmental health benefits, omitting meat or meat products has also been associated with a risk of being deficient in specific nutrients, such as iron. As data on the iron status among Norwegian vegans, vegetarians and pescatarians are lacking, the present study aimed to assess iron status in these groups of healthy adults. Blood markers for iron status were measured in 191 participants (18–60 years old) comprising 106 vegans, 54 vegetarians and 31 pescatarians: serum-ferritin (S-Fe), serum-iron (S-Iron) and serum-total iron binding capacity (S-TIBC). Serum-transferrin-saturation (S-TSAT) was estimated (S-Iron/S-TIBC × 100). The median concentration of blood markers for iron status were within the normal range with no difference between the different dietary practices. In total, 9% reported iron supplement use the last 24 h. S-Fe concentrations below reference (<15 μg/L) were found in 8% of the participating women, of which one participant reported iron supplement use. In multiple regression analysis, duration of adherence to dietary practices and the female gender were found to be the strongest predictors for decreased S-Fe concentration. In conclusion, although the participants were eating a plant-based diet, the majority had sufficient iron status. Female vegans and vegetarians of reproductive age are at risk of low iron status and should have their iron status monitored.

## 1. Introduction

The current food systems and evolving dietary patterns with increasing consumption of animal products and processed foods have an environmental impact as well as implications leading to global malnutrition [1,2,3,4,5]. A reversal of these trends towards plant-based diets has been widely recommended to reduce the environmental impact of food systems and to improve population health [6,7,8,9,10]. To date there is no formal definition of ‘vegetarian’ and ‘vegan’, but, in short: a vegetarian is someone who does not eat any meat, poultry, fish, shellfish or by-products of animal slaughter; a vegan is a vegetarian who avoids all animal and animal-derived products. 

Vegetarian and vegan diets have specifically been associated with a lower prevalence of type 2 diabetes, a lower incidence and/or mortality of ischemic heart disease, lower odds of hypertension and a lower incidence from total cancer [5,6,7,11]. On the other hand, vegetarians and vegans have been identified as groups at risk of micronutrient deficiency, and iron is a micronutrient of concern related to strict plant-based diets [12,13,14,15]. There are two major forms of iron found in food: heme iron, which is only found in animal products, and non-heme iron, which is found in both plant foods and animal products [16]. Heme iron is well absorbed (13–35%), whereas absorption of non-heme iron is much lower (2–20%) and depends on the iron status and dietary inhibitors, such as phytic acid, polyphenols, calcium and peptides (from partially digested proteins), and enhancers such as ascorbic acid and muscle tissue [17]. Phytate is one of the most potent absorption inhibitors and is found in whole grains, legumes and nuts. More than 50% of phytate intake comes from grain products, which is significant because grain products are also the most significant source of dietary iron for many vegetarians [18]. Vitamin C, and/or other organic acids, enhances the absorption of iron. According to a national dietary survey in Norway, the average iron intake ranges from 10 to 13 mg/day, with lower iron intakes in women compared to men [19].

Iron is essential for multiple biological functions in humans, including the synthesis of heme compounds (hemoglobin and myoglobin) responsible for oxygen transport as well as the formation of heme enzymes and other iron-containing enzymes involved in electron transfer and oxidation-reductions [18,20]. Moreover, iron is involved in the functioning of the immune system and cognition [21,22]. Iron deficiency may exist with or without anemia [23], and even mild and moderate forms of iron deficiency may have severe health consequences characterized by gastrointestinal disturbances [24,25], impaired cognitive function [26,27], impaired immune function [28], fatigue, and decreased work performance, which is most likely due to reduced oxygen transport associated with anemia [29].

Nutritional iron deficiency is a consequence of low iron supply or poor dietary absorption [15,17,30]. No special dietary recommendations are given for vegans, vegetarians or pescatarians in Norway since a sufficient iron intake may be obtained on a well-balanced diet omitting meat and meat products [13,31,32,33]. However, an analytic review and a systematic review, and meta-analysis, reported that vegetarians and vegans are more likely to have low iron stores, iron depletion and associated iron deficiency anemia compared to non-vegetarians [14,15]. Despite lower iron stores among vegetarian adults than non-vegetarians, their S-Fe levels are usually within the normal range [34,35]. In previous studies, researchers have emphasized the importance of taking multiple measures of iron status into consideration when examining the prevalence of iron deficiency and eventually anemia [36,37]. 

Thus, the main aim of the present study was to examine iron status and the prevalence of iron deficiency using the blood markers S-Fe, S-Iron, S-TIBC, and S-TSAT among vegan, vegetarian and pescatarian adults in Norway. 

## 2. Materials and Methods

### 2.1. Subjects

From September to November 2019, we recruited 205 healthy participants—115 vegans, 55 vegetarians and 35 pescatarians—from the Oslo and Viken area in Norway. The primary endpoint of the study was iodine and B12 status [38]; however, we were also able to measure iron status. The participants were recruited through social media with convenience sampling and snowball sampling methods. The participants were recruited through closed Facebook groups and in online vegan and vegetarian forums. Inclusion criteria were as follows: (1) having a strict vegan, vegetarian or pescatarian diet for six months or more; and (2) participants had to be 18 years or older. Participants who reported use of thyroid medication or consumption of meat-based products were excluded. For more detailed information, see the previously published study on iodine status in vegans, vegetarians and pescatarians [38]. Additionally, participants who had missing biomarkers of S-Fe were excluded from all descriptive analyses (*n* = 14). After provided consent, participants answered an electronic questionnaire assessing background information (age, height and weight, marital status, level of education, smoking habits, country of birth, language, etc.). 

### 2.2. Analysis of Serum-Ferritin (S-Fe), Serum-Iron, S-Total Iron Binding Capacity (S-TIBC) and Serum-Transferrin Saturation (S-TSAT) in Blood

Serum concentration of S-Fe, S-Iron and S-TIBC were measured, and S-TSAT were estimated (S-Iron/S-TIBC × 100) to evaluate iron status. The participants provided a non-fasting blood sample. Blood for serum analysis (S-Fe, S-TIBC, S-Iron, S-TSAT) was collected in a 5.0 mL tube (BD vacutainer SST II advance, Becton Dickinson, Franklin Lakes, NJ, USA). The tubes were mixed gently by five inversions and placed in a rack. The blood samples were stored at room temperature before centrifugation at 1500 rpm for 10 min (centrifuge model 5804, Eppendorf, Hamburg, Germany). The samples were centrifuged after 30 to 120 min, then separated immediately. Thereafter, all samples were refrigerated (4 °C) until analysis within three days at the Fürst medical laboratory (Oslo, Norway). The assays were performed with ADVIA Centaur XP System and XPT System by immunoassays using chemiluminescence technology, Erlangen, Germany. 

### 2.3. Assessment of Iron Supplement Use

Iron supplement use was assessed by answering yes/no if iron supplements had been consumed within the last 24 h. The participants thereafter self-reported the amount of iron consumed in the supplement. 

### 2.4. Definition

In this study, participants were categorized as vegan if intakes of milk/milk products, fish, eggs, meat/meat products or poultry were reported as never, and as vegetarian if no intake of meat/meat products, poultry or fish was reported, but if intake of milk/milk products or cheese or eggs was reported. The participants were categorized as pescatarians if intakes of fish were reported and if no intake of meat/meat product or poultry were reported. 

To evaluate iron status, we used the reference values provided by Fürst medical laboratory (Oslo, Norway). S-Fe was considered sufficient in the range 15–200 μg/L in female participants and in the range 20–300 μg/L in male participants. S-Iron was considered sufficient in the range 9–34 μmol/L. S-TIBC was considered adequate in the range 49–83 μmol/L. S-TSAT was considered adequate in the range 10–50% in females (0–49 years), in the range 15–50% for females (>50 years) and in the range 15–57% for males.

### 2.5. Statistics

IBM SPSS version 27 (IBM Corp., Armonk, NY, USA) was used for statistical analysis. The distribution of the data was checked using normality tests. We presented normally distributed data as mean ± standard deviation (SD) and non-normally distributed data as median with the 25th and 75th percentiles (p25, p75). We used the Kruskal Wallis test to test for differences between the continuous non-parametric variables age, body mass index (kg/m^2^) (BMI), S-Fe, S-TIBC, S-Iron and S-TSAT concentration with their dietary practice (vegan, vegetarian or pescatarian), and the *p*-value < 0.05 was used as the significance level. The Chi-square test was used to test for differences using categorical variables (Table 1). 

Multiple linear regressions were used to evaluate the association between the continuous non-parametric variable S-Fe and different independent variables. The dependent variable S-Fe was skewed and was therefore log−10 transformed prior to the regression analysis. Simple regressions were performed prior to the multiple regression analysis to examine the following covariates: gender, body mass index (kg/m^2^), age, educational level, pregnancy, lactation, dietary practice, duration of adherence to their dietary practice, 24 h use of iron supplements, and smoking. Independent variables that were significantly associated (*p* < 0.05) with the dependent variable S-Fe were included in a preliminary multiple regression model. Variables were excluded if not significant (*p* < 0.05). Only significant variables were retained (gender, duration of adherence to dietary practice) in the final regression model. Thereafter, analysis of the residuals was performed to examine the fit of the model. 

## 3. Results

Table l below presents the background characteristics of the vegans (*n* = 106), vegetarians (*n* = 54) and pescatarians (*n* = 31). Mean adherence to vegan, vegetarian or pescatarian diet was 5 ± 3 years, with the range 1–10 years. There was a significant difference in the length of adherence between the dietary practices, with the lowest adherence in the vegan group and the longest adherence in the pescatarian group. A total of 17 participants reported to have used iron supplements in the last 24 h, of which 15 participants reported the amount of iron ingested. Among those reporting the use of iron supplements during the last 24 h, 12 participants were female vegans, and 5 participants were male vegetarians. The median (p25, p75) iron intake from supplements was 26 (15, 28) µg/day, with a range of 7–101 µg. 

### 3.1. Blood Markers of Iron Status

The median concentrations of S-Fe, S-Iron and S-TIBC, and the estimated S-TSAT, are presented below in Table 2 and Table 3. All of the median blood markers of iron status—S-Fe, S-Iron, S-TIBC, and S-TSAT—were within the reference range for females (Table 2) and males (Table 3). Among all the females (vegans, vegetarians and pescatarians), 8% had S-Fe concentrations below reference (<15 μg/L). Of these, one participant reported 24 h iron supplement use, half of these (*n* = 8) were below 30 years of age and nearly half were vegans (*n* = 7). No men had S-Fe concentrations below reference values (<20 μg/L). Concentrations above reference values (200 µg/L females; 300 µg/L males) were found in 5% of males, but not in female participants. The female participants had lower S-Fe concentrations compared to the male participants; vegan females vs. men (*p* ≤ 0.001), vegetarian females vs. men (*p* ≤ 0.001) and pescatarian females vs. men (*p* = 0.039) (Figure 1).

### 3.2. Predictors for Serum-Ferritin Concentration

The predictors of LogS-Fe found in multiple linear regression models are presented in Table 4. The female gender was found to be associated with decreased S-Fe concentration, compared to the male gender. A longer duration of adherence to vegan, vegetarian or pescatarian dietary practice was also found to be associated with decreased S-Fe concentration. These predictors explained 32% of the variance found in S-Fe concentration (*p* ≤ 0.001). 

## 4. Discussion

To our knowledge, this present study is among the first to include multiple measures to evaluate iron status among Norwegian vegans, vegetarians and pescatarians. Although the participants were eating restricted plant-based diets, the majority had sufficient iron status evident by blood markers within the reference range in multiple measures (S-Fe, S-Iron, S-TIBC, S-TSAT). No difference was found in iron status between the dietary groups. 

### 4.1. Iron Status and Plant-Based Dietary Practice

Low S-Fe concentration is regarded as a sensitive indicator of iron deficiency in healthy subjects with no known inflammation [39], and several studies have revealed a lower concentration of S-Fe among vegetarians, compared to non-vegetarians [14,15,40]. However, these studies used different cut off criteria, methods, definitions of vegetarian diet, inclusions criteria, and one study was based on runners [40]. In this present study, 8% of the female participants (including vegans, vegetarians and pescatarians) were classified as iron deficient (S-Fe < 15 µg/L). Elevated S-Fe (>300 μg/L) were only present in male participants (5%) suggesting possible iron overload. Results from the present study indicate a significantly higher prevalence of low S-Fe among females in all dietary groups compared to men, which is consistent with findings in populations with non-vegetarian [41] and vegetarian diets [14,15,40,42]. S-Fe in the present study was comparable to studies on vegetarians and non-vegetarians in Switzerland (11 and 14 µg/L) [43], Germany (60 and 69 ng/mL) [44] and Finland (26 and 72 µg/L) [45]. However, these studies did not present disaggregated data for men and women. In the present study, no differences were found in dietary practice and S-Fe for women; however, in men, vegans had significantly lower S-Fe than vegetarians. In the study from Switzerland, iron intake was positively correlated with P-Fe in non-vegetarians and vegetarians, but not vegans. P-Fe was significantly higher in non-vegetarians, despite a higher iron intake in the vegans [43]. In the study from Germany [44] vegans and non-vegetarians had a similar iron status. In the study from Finland, the vegans had a significantly lower P-Fe [45]. These studies support the fact that vegan diets may compromise iron status, as they provide non-heme iron which has a low bioavailability, and also a high consumption of iron absorption inhibitors [15,43].

In this present study, gender and duration of adherence to vegan or vegetarian dietary practice explained 32% of the variance found in S-Fe. S-Iron is another marker for evaluation of iron status; however, it is not regarded as a good indicator to evaluate the depletion of iron stores. S-Iron is not a sensitive measure of iron status, partly because of daily fluctuations. For enhanced utility, serum iron measurements are used in conjunction with S-TIBC measurements. We found male and female participants to have similar levels of S-Iron and S-TIBC in all dietary groups, and that the median levels were within the normal range. Similar findings were reported in Spanish [42] and German [40] studies with vegans and lacto-ovo vegetarians. In our study, neither S-Iron nor S-TIBC was associated with the length of adherence to dietary practice or gender, similar to the findings reported in the German study [40]. Another indicator that provides information about the adequacy of iron supply is S-TSAT, the ratio of S-Iron to S-TIBC. None of the participants in the present study had S-TSAT below reference, supporting the assumption of sufficient iron status at group level. Our data suggest that even on a vegan diet it is possible for both genders to obtain adequate iron status as measured by these blood markers.

### 4.2. Iron Supplement Use

In this study, the reported use of iron supplements was low (9%), of which 12 participants were female vegans and 5 participants were male vegetarians. However, supplement use in our study is similar to the low use of iron supplements reported among female (6% took iron supplements every day) and male (4% did the same) Norwegian students [46]. Low iron supplement use was also reported in the German study among vegan and lacto-ovo vegetarians, with fewer than one-fifth of the participants being iron supplement users [40]. In our study, we did not find iron supplement use to be a predictor of increased iron status, possibly due to the reported low usage of iron supplements and the use of 24 h iron supplements instead of habitual usage. If iron supplement use had been assessed by habitual use, we might have seen an association between iron status and iron supplement use. 

### 4.3. Further Research

It is unclear which blood-based marker best reflects the iron status in vegans, vegetarians and pescatarians. Data are scarce on blood markers for evaluation of iron status, other than hemoglobin and S-Fe in vegetarians [15], and in vegans data are even more limited. In our study, biochemical iron status was assessed using blood serum. Further research should evaluate red blood cells in young vegans and vegetarians, especially young females. Future studies should also explore the impact of determinators such as dietary enhancers, dietary inhibitors, iron supplement usage and studies compared with a control group. 

### 4.4. Strengths and Limitations

The present study has some limitations and several strengths. The blood markers for iron status were analyzed by standard methods in a recognized Norwegian laboratory (Fürst). Unfortunately, we were not able to measure hemoglobin in this study, and iron-deficiency anemia [39] could therefore not be estimated by blood markers. However, the use of iron supplements was reported to be overall low. The use of iron supplements may have been underestimated due to the use of a 24 h recall method. Disadvantages of this method include the inability of a single day’s intake to describe habitual supplemental use. In addition, the primary outcome of this study was not iron status, but iodine and B12 status. Therefore, the participants were not asked specifically about iron supplements, but about supplement use in general, and the consequence might be insufficient data on iron supplement use. 

Most of the participants had normal weight and were non-smokers, indicating a generally health-conscious cohort. At inclusion, the participants had been following their diets for several years, and probably their dietary and supplement habits were stable. Consequently, the participants in our study may have experience and knowledge on how to prepare a nutrient-dense plant-based diet. These participants were students and mainly recruited through social media, and presumably their nutritional knowledge and health consciousness was high. Self-selection bias may threaten internal validity, but the diversity (gender, age range) in our sample enhances the robustness of the findings. The major limitation is the lack of an age- and sex-matched control group. Further, the numbers of males were less than half the number of females, and the number of pescatarians was also low. However, the number of vegans was twice that of the vegetarians, which gives strength to the data since the vegans have the most restricted diet. Another limitation is that the participants were not asked to undergo a wash out period (iron supplements, oral contraceptives or use of medication that may interfere with iron status) ahead of participation in the study. Recruitment would have been even more difficult if we had asked the participants to change their habitual living habits, such as their use of medications and oral contraceptives. 

## 5. Conclusions

The majority of the vegans, vegetarians and pescatarians in the Oslo area in Norway had sufficient iron status. Female vegans and vegetarians of reproductive age might be at risk of low iron status as women of fertile age have increased needs for iron because of losses due to menstrual bleeding. Young women with restrictive diets should have their iron status monitored. 

## Figures and Tables

**Figure 1 biomolecules-11-00454-f001:**
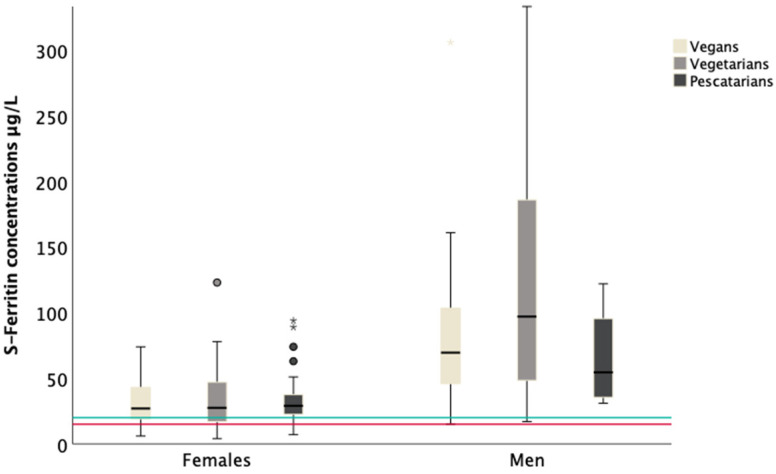
S-Fe concentration (μg/L) is presented according to gender and dietary practice. Vegan females (*n* = 66), vegetarian females (*n* = 42), pescatarian females (*n* = 27), vegan men (*n* = 40), vegetarian men (*n* = 12), pescatarian men (*n* = 4). Box plot details are as follows: the horizontal black lines indicate the median, the boxes indicate the interquartile range (IQR) (25th percentile to 75th percentile), the whiskers represent observations within 1.5 times the IQR. Outliers 1.5 times away from the IQR are marked as circles. Extreme outliers are marked as stars, and extreme outliers above 350 μg/L are excluded from the figure. The marked red line shows the cut-off for iron deficiency in females (<15 μg/L), the marked green line shows the cut- off for iron deficiency in men (<20 μg/L). The cut-offs used are based on recommended S-Fe levels by Fürst (Norwegian laboratory).

**Table 1 biomolecules-11-00454-t001:** Background characteristics of vegans, vegetarians and pescatarians in Norway (*n* = 191).

	Vegans n (%)	Vegetarians n (%)	Pescatarians n (%)	Total n (%)	*p*-Value ^a^
Participants	106	54	31	191	
Age ^b^	31 ± 9 (19–56)	30 ± 10 (18–60)	29 ± 8 (20–52)	31 ± 9 (18–60)	0.167
BMI, kg/m^2 b^	23 ± 3 (17–33)	23 ± 4 (18–40)	23 ± 3 (18–32)	23 ± 3 (17–40)	0.836
Duration dietary practice (years) ^b^	4 ± 3 (1–10)	5 ± 3 (1–10)	6 ± 4 (1–10)	5 ± 3 (1–10)	0.030 *
Gender Females Males	66 (62) 40 (38)	42 (78) 12 (22)	27 (87) 4 (13)	135 (71) 56 (29)	0.011 *
Previously given birth ^c^ Yes No Number of children ^b^	9 (8) 57 (54) 2 ± 1 (1–4)	7 (13) 35 (65) 2 ±1 (1–4)	5 (16) 22 (71) 2 ± 1 (1–5)	114 (60) 21 (11) 2 ± 1 (1–5)	0.817 0.839
Planning pregnancy ^d^ Yes No Currently pregnant	13 (12) 52 (49) 1 (1)	5 (9) 37 (69) 0	6 (19) 21 (68) 0	24 (13) 110 (58) 1 (1)	0.622
Country of origin Norway Other countries	88 (83) 18 (17)	44 (82) 10 (18)	25 (81) 6 (19)	157 (82) 34 (18)	0.515
Educational level <12 years 12 years 1–4 years higher education	2 (2) 18 (17) 86 (81)	1 (2) 11 (20) 42 (78)	2 (7) 2 (7) 27 (87)	5 (3) 31 (16) 155 (81)	0.317
Smoking status No Yes	94 (89) 12 (11)	48 (89) 6 (11)	29 (94) 2 (6)	171 (90) 20 (10)	0.876
Iron supplement, 24 h ^e^ Yes No	12 (11) 94 (89)	2 (4) 52 (96)	3 (10) 28 (90)	17 (9) 174 (91)	0.274

^a^*p*-values for difference between continuous variables (Kruskal Wallis test); (Chi-Square test for difference with categorical variables); ^b^ Mean values ± SD (min–max); ^c^ Have you previously given birth, yes/no; ^d^ Are you planning a pregnancy the next two years?; ^e^ Iron supplement use, assessed by 24 h recall; * Significant level used < 0.05.

**Table 2 biomolecules-11-00454-t002:** Concentrations of measured non-fasting blood markers for iron status in female vegans (*n* = 66), vegetarians (*n* = 42) and pescatarians (*n* = 27).

Blood Markers	Vegans ^a^	n	Vegetarians ^a^	n	Pescatarians ^a^	n	*p*-Value ^b^	Reference Value
S-Fe	27 (19, 43)	66	28 (17, 48)	42	29 (22, 38)	27	0.803	15–200 µg/L
S-Iron	19 (13, 23)	65	18 (13, 23)	42	19 (13, 21)	26	0.878	9–34 µmol/L
S-TIBC	69 (63, 76)	66	71 (63, 78)	42	69 (64, 79)	27	0.649	49–83 µmol/L
S-TSAT	26 (19, 33)	65	26 (19, 35)	42	26 (21, 34)	26	0.942	Female 0–49 years, 10–50 % Female ≥ 50 years, 15–50%

^a^ Values are presented as median (p25, p75), ^b^ Test for difference (Kruskal Wallis test), *p*-value < 0.05 was used as significance level.

**Table 3 biomolecules-11-00454-t003:** Concentrations of measured non-fasting blood markers for iron status in male vegans (*n* = 40), vegetarians (*n* = 12) and pescatarians (*n* = 4).

Blood Markers	Vegans ^a^	n	Vegetarians ^a^	n	Pescatarians ^a^	n	*p*-Value ^b^	Reference Value
S-Fe	70 (45, 104)	40	97 (47, 191)	12	55 (33, 109)	4	0.516	20–300 µg/L
S-Iron	19 (15, 23)	39	21 (18, 22)	12	17 (9, 23)	4	0.581	9- 34 µmol/L
S-TIBC	66 (62, 73)	40	68 (64, 70)	12	62 (58, 72)	4	0.347	49–83 µmol/L
S-TSAT	27 (20, 34)	39	31 (26, 34)	12	27 (14, 36)	4	0.620	15–57 %

^a^ Values are presented as median (p25, p75), ^b^ Test for difference (Kruskal Wallis test), *p*-value < 0.05 used as significance level.

**Table 4 biomolecules-11-00454-t004:** Predictors of serum-ferritin concentration in vegans, vegetarians and pescatarians (*n* = 191).

Dependent Variable	Predictor Variables	Unadjusted Coefficient (CI 95%)	*p*-Value	Adjusted Coefficient (CI 95%)	*p*-Value
LogS-Fe ^1^					
	Duration of dietary practice ^2^	−0.2 (−0.3, −0.1)	0.003	−0.2 (−0.1, −0.0)	<0.001
	Gender ^3^	0.4 (0.3, 0.5)	<0.001	0.5 (0.1, 0.2)	<0.001

^1^ Log-transformed serum-ferritin concentrations; ^2^ Duration of vegan/vegetarian/pescatarian practice, continuous variable ranging from 1 to 10 years; ^3^ Gender (0 = female, 1 = male).

## Data Availability

The data presented in this study are available on request from the corresponding author.

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
