# Peer review of "Iron Status of Vegans, Vegetarians and Pescatarians in Norway"

_biomolecules, 2021, doi:10.3390/biom11030454_

Round 1
Reviewer 1 Report
This cross-sectional p study comprising 191 participants investigated the iron status in vegans, vegetarians and pescetarians from Norway.
The topic is in general very interesting and of public health relevance. However, I have a few comments that should be addressed.
Major comment:
Unfortunately, omnivores were not included in this study. The authors themselves mention this limitation, but could come up with other studies, which investigated the iron status in plant-based diets compared to omnivorous diet. These studies should be included and discussed in the discussion of the manuscript on the one hand as further studies investigating the iron status in plant-based diets and on the other hand to address the risk of iron deficiency in omnivores:
Vitamin and Mineral Status in a Vegan Diet.
-Micronutrient status and intake in omnivores, vegetarians and vegans in Switzerland.
Schüpbach R, Wegmüller R, Berguerand C, Bui M, Herter-Aeberli I. Eur J Nutr. 2017 Feb;56(1):283-293. doi: 10.1007/s00394-015-1079-7. Epub 2015 Oct 26.PMID: 26502280
-Weikert C, Trefflich I, Menzel J, Obeid R, Longree A, Dierkes J, Meyer K, Herter-Aeberli I, Mai K, Stangl GI, Müller SM, Schwerdtle T, Lampen A, Abraham K. Dtsch Arztebl Int. 2020 Aug 31;117(35-36):575-582. doi: 10.3238/arztebl.2020.0575.PMID:
-Food and Nutrient Intake and Nutritional Status of Finnish Vegans and Non-Vegetarians.
Elorinne AL, Alfthan G, Erlund I, Kivimäki H, Paju A, Salminen I, Turpeinen U, Voutilainen S, Laakso J. PLoS One. 2016 Feb 3;11(2):e0148235. doi: 10.1371/journal.pone.0148235. eCollection 2016.PMID: 26840251
Minor comments:
- The age range of participants should be mentioned in the abstract.
- In the introduction the authors may try to find references from original studies or systematic reviews.
- More information of inclusion and exclusion criteria are given in ref. 35, but may be of interest for the reader in this manuscript, too.
- In the table of characteristics “previously given birth” and “planning pregnancy” are shown – please give the definitions of these characteristics - why and what questions were asked to the participants.
- For reference 37 it should be mentioned in the text – that this study was based on runners.
This cross-sectional p study comprising 191 middle-aged healthy participants investigated the iron status in vegans, vegetarians and pescetarians from Norway.
The topic is in general very interesting and of public health relevance. However, I have a few comments that should be addressed.
Major comment:
Unfortunately, omnivores were not included in this study. The authors themselves mention this limitation, but could come up with other studies, which investigated the iron status in plant-based diets compared to omnivorous diet. These studies should be included and discussed in the discussion of the manuscript on the one hand as further studies investigating the iron status in plant-based diets and on the other hand to address the risk of iron deficiency in omnivores:
Vitamin and Mineral Status in a Vegan Diet.
-Micronutrient status and intake in omnivores, vegetarians and vegans in Switzerland.
Schüpbach R, Wegmüller R, Berguerand C, Bui M, Herter-Aeberli I. Eur J Nutr. 2017 Feb;56(1):283-293. doi: 10.1007/s00394-015-1079-7. Epub 2015 Oct 26.PMID: 26502280
-Weikert C, Trefflich I, Menzel J, Obeid R, Longree A, Dierkes J, Meyer K, Herter-Aeberli I, Mai K, Stangl GI, Müller SM, Schwerdtle T, Lampen A, Abraham K. Dtsch Arztebl Int. 2020 Aug 31;117(35-36):575-582. doi: 10.3238/arztebl.2020.0575.PMID:
-Food and Nutrient Intake and Nutritional Status of Finnish Vegans and Non-Vegetarians.
Elorinne AL, Alfthan G, Erlund I, Kivimäki H, Paju A, Salminen I, Turpeinen U, Voutilainen S, Laakso J. PLoS One. 2016 Feb 3;11(2):e0148235. doi: 10.1371/journal.pone.0148235. eCollection 2016.PMID: 26840251
Minor comments:
- The age range of participants should be mentioned in the abstract.
- In the introduction the authors may try to find references from original studies or systematic reviews.
- More information of inclusion and exclusion criteria are given in ref. 35, but may be of interest for the reader in this manuscript, too.
- In the table of characteristics “previously given birth” and “planning pregnancy” are shown – please give the definitions of these characteristics - why and what questions were asked to the participants.
- For reference 37 it should be mentioned in the text – that this study was based on runners.

Author Response
We thank the reviewers for many valuable inputs that have improved the manuscript.
Review Report (Reviewer 1)
Unfortunately, omnivores were not included in this study. The authors themselves mention this limitation, but could come up with other studies, which investigated the iron status in plant-based diets compared to omnivorous diet. These studies should be included and discussed in the discussion of the manuscript on the one hand as further studies investigating the iron status in plant-based diets and on the other hand to address the risk of iron deficiency in omnivores:
Vitamin and Mineral Status in a Vegan Diet.
-Micronutrient status and intake in omnivores, vegetarians and vegans in Switzerland.
Schüpbach R, Wegmüller R, Berguerand C, Bui M, Herter-Aeberli I. Eur J Nutr. 2017 Feb;56(1):283-293. doi: 10.1007/s00394-015-1079-7. Epub 2015 Oct 26.PMID: 26502280
-Weikert C, Trefflich I, Menzel J, Obeid R, Longree A, Dierkes J, Meyer K, Herter-Aeberli I, Mai K, Stangl GI, Müller SM, Schwerdtle T, Lampen A, Abraham K. Dtsch Arztebl Int. 2020 Aug 31;117(35-36):575-582. doi: 10.3238/arztebl.2020.0575.PMID:
-Food and Nutrient Intake and Nutritional Status of Finnish Vegans and Non-Vegetarians.
Elorinne AL, Alfthan G, Erlund I, Kivimäki H, Paju A, Salminen I, Turpeinen U, Voutilainen S, Laakso J. PLoS One. 2016 Feb 3;11(2):e0148235. doi: 10.1371/journal.pone.0148235. eCollection 2016.PMID: 26840251
Reply: Thank you for the comment, we agree and have added the suggested references and a discussion about iron status in plant-based diets compared to omnivorous diets, see line 226.
Minor comments:
- The age range of participants should be mentioned in the abstract.
Reply: We have revised this, please see line 14.
- In the introduction the authors may try to find references from original studies or systematic reviews.
Reply: We agree and have added references from original studies or systematic reviews, see references 6-10 and 12-15.
Craig WJ. Health effects of vegan diets. Review. Am J Clin Nutr. 2009 May;89(5):1627S-1633S
Aleksandrowicz, L., et al., The Impacts of Dietary Change on Greenhouse Gas Emissions, Land Use, Water Use, and Health: A Systematic Review. PLoS One, 2016. 11(11): p. e0165797
Dinu, M., et al., Vegetarian, vegan diets and multiple health outcomes: A systematic review with meta-analysis of observational studies. Crit Rev Food Sci Nutr, 2017. 57(17): p. 3640-3649
Haider, L.M., et al., The effect of vegetarian diets on iron status in adults: A systematic review and meta-analysis. Crit Rev Food Sci Nutr, 2018. 58(8): p. 1359-1374
Pawlak, R., J. Berger, and I. Hines, Iron Status of Vegetarian Adults: A Review of Literature. Am J Lifestyle Med, 2018. 12(6): p. 486-498
- More information of inclusion and exclusion criteria are given in ref. 35, but may be of interest for the reader in this manuscript, too.
Reply: We agree and have added more information about the inclusion and exclusion criteria in the revised manuscript, see lines 78-85.
- In the table of characteristics “previously given birth” and “planning pregnancy” are shown – please give the definitions of these characteristics - why and what questions were asked to the participants.
Reply: We agree and have included information in footnotes, see footnotes in Table 1.
- For reference 37 it should be mentioned in the text – that this study was based on runners.
Reply: We agree and have mentioned in the discussion that the study was based on runners,
see lines 226.
Reviewer 2 Report
The study is interesting and important considering that more and more people change their dietary (eating) habits and move to meatless diets. The experiments have been performed correctly. I have no any objection to methodology. Results are presented accurately.
However, the manuscript leaves the Reader unsatisfied. The characterization of biochemical iron status in the blood serum without the evaluation of red blood cell (RBC) status in parallel is of limited significance for the Readers. Changes in iron parameters could be particularly interesting in the context of resulting consequences on RBC status and possible diagnosis of an early stage of iron deficiency anemia (for example in female vegans and vegetarians in reproductive age). Authors themselves admit that the lack of hemoglobin concentration results is one of limitations of the study. Indeed, it is a serious limitation. The Authors’ explanation (practical reasons) is very general and somehow vague. It is diffcult to understand why the measurement of not only RBC (a classical, simple and routine analysis, requiring a tiny volume of blood) but also reticulocyte indices was not performed. Most of hematology analysers make it possible to predict the function of the red cell system on the basis of reticulocyte parameters. Mean reticulocyte haemoglobin content deserves special attention in this context. This parameter is commonly used in human medicine. The RBC/reticulocyte data could strongly increase the value of the study. I understand that it is no longer possible to make up these gaps but at least Authors should provide a broader discussion linking iron in the serum with possible RBC/reticulocyte status.
If possible, I strongly suggest performing the analysis of hepcidin end erythropoietin levels in the blood serum of volunteers from 3 experimental groups. The Authors still should have frozen serum samples. Appropriate ELISA assays are largely available. This could help combining iron data with RBC system.
Author Response
We thank the reviewers for many valuable inputs that have improved the manuscript.
Review Report (Reviewer 2)
This cross-sectional study comprising 191 middle-aged healthy participants investigated the iron status in vegans, vegetarians and pescetarians from Norway.
The topic is in general very interesting and of public health relevance. However, I have a few comments that should be addressed.
The study is interesting and important considering that more and more people change their dietary (eating) habits and move to meatless diets. The experiments have been performed correctly. I have no any objection to methodology. Results are presented accurately.
However, the manuscript leaves the Reader unsatisfied. The characterization of biochemical iron status in the blood serum without the evaluation of red blood cell (RBC) status in parallel is of limited significance for the Readers. Changes in iron parameters could be particularly interesting in the context of resulting consequences on RBC status and possible diagnosis of an early stage of iron deficiency anemia (for example in female vegans and vegetarians in reproductive age). Authors themselves admit that the lack of hemoglobin concentration results is one of limitations of the study. Indeed, it is a serious limitation. The Authors’ explanation (practical reasons) is very general and somehow vague. It is diffcult to understand why the measurement of not only RBC (a classical, simple and routine analysis, requiring a tiny volume of blood) but also reticulocyte indices was not performed. Most of hematology analysers make it possible to predict the function of the red cell system on the basis of reticulocyte parameters. Mean reticulocyte haemoglobin content deserves special attention in this context. This parameter is commonly used in human medicine. The RBC/reticulocyte data could strongly increase the value of the study. I understand that it is no longer possible to make up these gaps but at least Authors should provide a broader discussion linking iron in the serum with possible RBC/reticulocyte status.
Reply: We agree that evaluation of the red blood cells in relation to the biochemical analysis in serum would provide important data on the severity of the iron status. However, WHO recommend measurements of serum ferritin as the best indicator evaluating iron status at population level in places were infection diseases are not common, as in Norway (WHO, 2007). We have added a broader discussion linking iron in serum with possible RBC/reticulocyte status, as suggested, see lines 232-241 and in need for future research, see lines 271-273.
WHO 2007 Assessing Iron status of populations. Second edition Including Literature Reviews. https://apps.who.int/iris/bitstream/handle/10665/75368/9789241596107_eng.pdf;jsessionid=42C1E89B2056A8762A698D1B8CE4AC8C?sequence=1
If possible, I strongly suggest performing the analysis of hepcidin end erythropoietin levels in the blood serum of volunteers from 3 experimental groups. The Authors still should have frozen serum samples. Appropriate ELISA assays are largely available. This could help combining iron data with RBC system.
Reply: We agree that performing the analysis of hepcidin end erythropoietin levels in the blood serum, would be of great interest, but unfortunately, we don’t have the frozen serum samples. All biological samples were destroyed after the analysis were performed in the laboratory. In addition, the primary outcome of this study was not iron status, but iodine and B12 status. When we were able to measure iron status it was too late to measure hemoglobin, since hemoglobin should be measured within some hours after blood sampling, we have added information about this in lines 78-79.
Round 2
Reviewer 1 Report
The manuscript has been considerably improved. I agree with most changes.
But the added discussion line 235-241 is not sufficient. The following sentence "In contrast to the study from Switzerland nearly all the vegans and one-third of omnivoreshad consumed supplements the previous 4 weeks." is misleading. The reader may assume that all participants in the German study took iron supplements. The authors should more clearly discuss these studies.
Author Response
Thank you for making us aware of the error. We have updated the discussion, please see lines 234-245.
